# Observations on the Feeding of *Drymonema dalmatinum* in the Gulf of Trieste

**Saul Ciriaco** [1], **Lisa Faresi** [2] and **Marco Segarich** [3,*]

1   WWF Miramare MPA, Via Beirut 2-4, 34151 Trieste, Italy; saul@riservamarinamiramare.it
2   Agenzia Regionale per la Protezione dell'Ambiente della Regione Friuli Venezia Giulia, Via Cairoli 14, 33057 Palmanova, Italy; lisa.faresi@arpa.fvg.it
3   Shoreline Soc. Coop, Area Science Park, Loc. Padriciano 99, 34149 Trieste, Italy
*   Correspondence: marco.segarich@shoreline.it

**Keywords:** *Drymonema dalmatinum*; Gulf of Trieste; jellyfish; feeding behaviour; *Rhizostoma pulmo*; Miramare Marine Protected Area

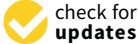



The largest scyphozoan jellyfish of the Mediterranean Sea, *Drymonema dalmatinum* was first described by Haeckel [1] from material collected off the Dalmatian coast of the Adriatic Sea. According to Malej [2], there is no information on *Drymonema* in the Adriatic from 1937 till 1984, when a diver photographed one individual in the small eastern Adriatic Bay of Žrnovnica. Since the year 2000, the number and frequency of sightings have increased slightly throughout the Adriatic Sea, but the species must still be considered rare in the region.

There are few documented sightings in the literature on the Mediterranean Sea in the last 10 years. According to Malej et al. [2], in the Adriatic Sea, there was one sighting in 2010 (Murter, HR) and two sightings in 2014 (Kotor bay, MNE; Gulf of Trieste, ITA). In the Mediterranean, there was a well-documented sighting in the Sea of Marmara in 2020 [3], and one sighting was reported along the Spanish coast in 2013 [4].

*Drymonema* belongs to the Cyaneidae family, which presents three valid genera—*Cyanea*, *Desmonema*, *Drymonema*—with about 20 valid species, and shows a considerable variance for the three characteristics diagnostics of the family. Although Haeckel in 1880 recognized the distinctiveness of *Drymonema*, coining the subfamily Drymonemidae, Bayha, and Dawson [5], he demonstrated the family Cyaneidae to be polyphyletic, and that *Drymonema* is clearly reciprocally monophyletic with respect to Cyaneaidae. Moreover, phylogeny and the high genetic differences highlight the need for assignment of a neotype of *D. dalmatinum*, in order to support the recognition of a new western Atlantic *Drymonema* species.

The umbrella is in the form of a flat disc, consisting of a thicker and more rigid central part and a thin, peripheric, marginal area without tentacles, with 20 lappets per octant. Four oral arms are very broad; have a large, curtain-like surface; and are nearly as long as the diameter of the bell. In larger specimens, there are clear brownish radial strakes on the exumbrellar surface.

A distinctive feature of the genus *Drymonema* is that large specimens feed primarily on *Aurelia* [6], which was taken up by Bayha and Dawson [4], who supposed an external digestion by protease given the reduction of gastric filaments. From the present observation, it can be assumed that *Drymonema* does not only prey on Aurelia, but also other jellyfish, even larger ones like *Rhizostoma*. According to this hypothesis, the increase in the number of sightings of *Drymonema* could be linked not only to the presence of *Aurelia*, but also to the regular presence of *Rhizostoma* in the Gulf of Trieste throughout the year.

On 1 June 2020, researchers from the Miramare Marine Protected Area performed monitoring activity in the areas adjacent to the Miramare MPA, and made an unusual sighting: a specimen of *Drymonema dalmatinum* (Figure 1). The underwater visibility was about 10 m, and a large number of jellies, including *Rhizostoma pulmo*, were present along

the column. The specimen was found at a depth of about 10 m (in a 14 m seabed), and the sea temperature was around 20 °C. In his oral arms were two prey of the species *Rhizostoma pulmo*. The organism was not collected, due to the regulations on monitoring activities for Miramare MPA staff. The planned monitoring activity did not involve the use of measuring instruments, but the researchers were able to estimate the approximate size thanks to the experience gained during the visual fish census activities they usually carry out for the MPA. The bell diameter was approximately 40 cm, as were the length of oral arms.

Video is available here: https://youtu.be/HZUmpDXytp4 (accessed on 2 March 2021).

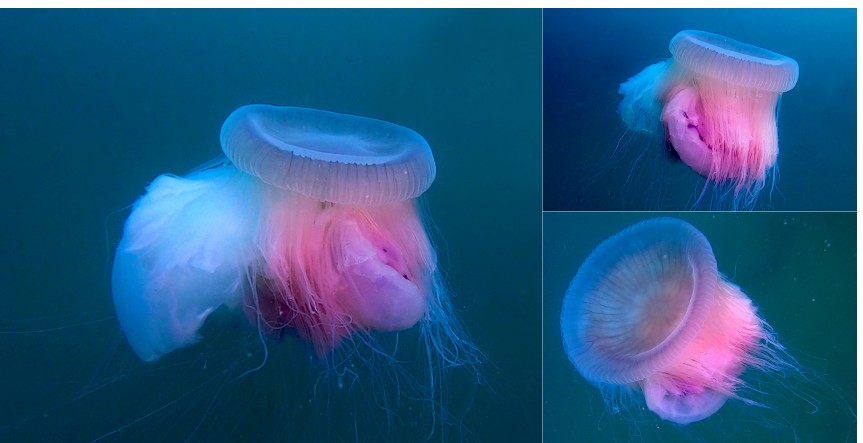

**Figure 1.** *Drymonema dalmatinum* preying on two *Rhizostoma pulmo* medusae.

**Author Contributions:** Conceptualization, S.C. and M.S.; software, S.C.; writing—original draft preparation, M.S., L.F. and S.C.; writing—review and editing, S.C. and M.S.; supervision, M.S. All authors have read and agreed to the published version of the manuscript.

**Funding:** This research received no external funding.

**Institutional Review Board Statement:** Not applicable.

**Informed Consent Statement:** Not applicable.

**Data Availability Statement:** The data presented in this study are available on request from the corresponding author.

**Conflicts of Interest:** The authors declare no conflict of interest.

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
