# Peer review of "Observations on the Feeding of Drymonema dalmatinum in the Gulf of Trieste"

_diversity, doi:10.3390/d13040163_

Round 1

Reviewer 1 Report

Dear editor and authors

I like these types of short submissions and I believe that they provide a valuable record of sightings and are worth publishing and sharing, rather than remaining hidden and forgotten on a hard drive somewhere. The video is really cool, and the article should be published.

The manuscript is generally well written, however, the sentence in lines 23 - 24 is difficult to understand, I am not sure what the authors are trying to say here, so that needs to be rephrased and clarified.

I think the authors could have delved into the literature a little more and included some further relevant material. There is a published sighting of D. dalmatinum of the Spanish coast during 2013 (Kienberger & Prieto, 2018), in the Sea of Marmara during 2020 (Öztürk 2020). I realise it’s not the Adriatic Sea, however, other sightings in the Mediterranean do help to build up a picture of the recent history of sightings. Finally, a sentence or two on the taxonomy/phylogenetics (Bayha & Dawson, 2010) of the species would be worth inclusion, I think. You can include up to 10 references, why not use that allowance to provide a slightly more detail. I think your potential readers will appreciate that, I certainly would. 

Bayha, K.M. and Dawson, M.N., 2010. New family of allomorphic jellyfishes, Drymonematidae (Scyphozoa, Discomedusae), emphasizes evolution in the functional morphology and trophic ecology of gelatinous zooplankton. The Biological Bulletin, 219(3), pp.249-267.

Kienberger, K. and Prieto, L., 2018. The first record of Drymonema dalmatinum from the northern Alboran Sea (western Mediterranean). Marine Biodiversity, 48(3), pp.1281-1282.

Öztürk, İ.D., 2020. The first record of Drymonema sp. from the Sea of Marmara, Turkey. Journal of the Black Sea/Medit Environ, 26(2), pp.231-237.

Well done on a nice discovery, I am quite jealous that you get to dive in nice warm clear water!

Kind regards

Author Response

Point 1:

The manuscript is generally well written, however, the sentence in lines 23 - 24 is difficult to understand, I am not sure what the authors are trying to say here, so that needs to be rephrased and clarified.

Response 1:

The original sentence was: Despite to the relative frequency of sightings starting from 2000 in the northern, middle and southern Adriatic, the paucity of this species records since its first record, confirms its rarity in the region.

We’ve hopefully clarified changing with this one:

“Since the year 2000, the number and frequency of sightings have increased slightly throughout the Adriatic Sea, but the species must still be considered rare in the region.” (LINES 18-20)

Point 2:

I think the authors could have delved into the literature a little more and included some further relevant material. There is a published sighting of D. dalmatinum of the Spanish coast during 2013 (Kienberger & Prieto, 2018), in the Sea of Marmara during 2020 (Öztürk 2020). I realise it’s not the Adriatic Sea, however, other sightings in the Mediterranean do help to build up a picture of the recent history of sightings.

Response 2:

We added few sentences with a short description of the last 10 year of sighting of Drymonema: LINES 22-26

Point 3:

Finally, a sentence or two on the taxonomy/phylogenetics (Bayha & Dawson, 2010) of the species would be worth inclusion, I think. You can include up to 10 references, why not use that allowance to provide a slightly more detail. I think your potential readers will appreciate that, I certainly would. 

Response 3: We’ve included recommended citations. Both for the sightings and for the genetic and morphological aspects (Line 27-40), in particular reconnecting also to the request of the reviewers 2&3

Reviewer 2 Report

I wrote my comments in the word document enclosed.

Author Response

Point 1:

The authors present an exceptional observation of Drymonema dalmatinum in the northern

Adriatic (Gulf of Trieste ,Italy). The importance this observation is not only due to the rarity

of this large jellyfish but to the fact that the authors were able to make also an amazing

movie of Drymonema eating two scyphomedusae of the species Rhizostoma pulmo. In my

knowledge, this is the first evidence that D. dalmatinum eats medusae of the genus

Rhizostoma and the authors must emphasize this aspect.

 I suggest them to reduce the morphological description and to implement the discussion, in particular referring to the study of Bayha and Dawson (“New Family of Allomorphic Jellyfishes, Drymonematidae (Scyphozoa, Discomedusae), Emphasizes Evolution in the Functional Morphology and Trophic Ecology of Gelatinous Zooplankton”, Biol. Bull 2010, 219:249-267). The observation presented in this manuscript, support the idea that larger Drymonema eat primarily large jellyfish digesting the medusae with proteases secreted by the enveloping oral arms and therefore justifying the absence of gastric filaments in larger specimens of this species.

Response 1: We followed suggestions and included morphogenetic citations on the uniqueness of the genus Drymonema, emphasizing predation on other jellyfish (lines 41-48)

Point 2:

Moreover, the authors could make the hypothesis that the increasing observation of

Drymonema since 2000 in the Adriatic could be linked also to an increase in the same area of

Rhyzostoma (a jellyfish that until now was not considered a prey for Drymonema).

Response 2: We also emphasized the uniqueness of predation on larger jellyfish in the genus Aurelia. LINES 41-48

Point 3:

Finally, I think that the picture is beautiful but not vey explanatory of the observed trophic

interaction: I suggest the authors to add also new images in a plate capturing some pictures

from their movie to better show the interaction between Drymonema and Rhyzostoma. I give

here some example of possible images:

Minor revisions:

- Give complete address of the authors

- Write in italic the genus of jellyfish

- Add information about how did you measure morphometry of Drymonema and its

prey. If these measurements are estimations, please write in the text as “The bell

diameter was around 40 cm….

Response 3: we added new images and hopefully fixed all errors and form

Reviewer 3 Report

Minor changes:

Line 31 there are two periods "40 cm.."

In line 32 "The organism was not collected due to the regulation of monitoring activities for Miramare MPA staff." It was a pity not to collect, at least, a small bit for genetic analysis (alcohol preserved). Have you seen more individuals of this species?

Line 34 "they were able to measure morphometry [which variables were measured?] and observe the species of prey organisms". The second part of the sentence should be written correctly, for example, "observe the specimen preying on its prey"

Author Response

Point 1:

Line 31 there are two periods "40 cm.."

Response 1: we fixed the sentence

Point 2:

In line 32 "The organism was not collected due to the regulation of monitoring activities for Miramare MPA staff." It was a pity not to collect, at least, a small bit for genetic analysis (alcohol preserved). Have you seen more individuals of this species?

Response2: No, we notice only that individuals. Yes, was a pity, we were not prepared for sampling jelly without collect entire individuals.

Point 3:

Line 34 "they were able to measure morphometry [which variables were measured?] and observe the species of prey organisms". The second part of the sentence should be written correctly, for example, "observe the specimen preying on its prey"

Response 3: We change the sentences (LINES 53-54)

Reviewer 4 Report

An interesting observation that deserves to be published. However, the text needs some editing (both grammar and spelling).

The species in question is erratic and unpredictable in appearance, and is usually seen in ones or twos. Field data are rare for this species. While we know that species of this genus eat other jellyfish from studies elsewhere, knowledge of what exactly is eaten by the different species is scant. Here the authors report on a prey species that has not previously been documented. Hence, while it is a novel observation it adds only incrementally to our understanding of the species' ecology. In my opinion, the paper does what it sets out to do, but the language needs to be tightened.

Author Response

Point 1:

In my opinion, the paper does what it sets out to do, but the language needs to be tightened.

Response 1: we’ve changed some sentence to fix it